# The Lipidomic Signature of Glioblastoma: A Promising Frontier in Cancer Research

**DOI:** 10.3390/cancers16061089

**Published:** 2024-03-08

**Authors:** Nina Yu, Orwa Aboud

**Affiliations:** 1School of Medicine, University of California, Davis, Sacramento, CA 95817, USA; 2Department of Neurology and Neurological Surgery, Comprehensive Cancer Center, University of California, Davis, Sacramento, CA 95817, USA

**Keywords:** lipidomics, glioblastoma, brain cancer

## Abstract

**Simple Summary:**

Glioblastoma is an aggressive brain cancer. Despite progress made in cancer research, patients diagnosed with glioblastomas continue to have a poor rate of survival. Lipidomics is a field of study that seeks to comprehend the structure and function of lipids, otherwise known as fats. Glioblastomas have demonstrated imbalance in lipids, and, therefore, lipidomics shows promise as an avenue to better understand glioblastomas. This paper reviews the literature on lipidomics in glioblastomas and suggests future directions for lipidomic research into the diagnosis, prognosis, and treatment of glioblastomas.

**Abstract:**

Glioblastoma is the most aggressive primary brain malignancy in adults, and has a survival duration of approximately 15 months. First line treatment involves surgical resection, chemotherapy, and radiation, but despite the multi-pronged approach and advances in cancer research, glioblastoma remains devastating with a high mortality rate. Lipidomics is an emerging discipline that studies lipid pathways and characteristics, and is a promising field to understand biochemical mechanisms. In glioblastoma, disrupted lipid homeostasis has been reported in the literature. A thorough understanding of serum lipidomics may offer ways to better understand glioblastoma biomarkers, prognosis, and treatment options. Here, we review the literature, offering future directions for lipidomics research in glioblastomas.

## 1. Introduction

Glioblastoma, otherwise known as World Health Organization (WHO) grade 4 astrocytoma, is an aggressive primary brain malignancy that often results in mortality within two years of diagnosis, despite aggressive treatment. The standard therapeutic regimen entails surgical debulking, chemotherapy using temozolomide (TMZ), and radiation therapy [1]. Even with this rigorous approach, glioblastoma adeptly mutates and quickly becomes resistant to treatment [2].

Though decades of research have been invested into understanding glioblastoma pathology, prognosis, and treatment, it remains a devastating diagnosis. The median survival of glioblastoma consistently hovers around 15 months [3]. Lipidomics as a field of study has demonstrated promise in understanding glioblastoma pathology and prognosis, and as a method to develop therapies. The mention of lipidomics first surfaced in the literature in the early 2000s, coinciding with advancements in electrospray ionization mass spectrometry [4,5]. Lipidomics quickly became a formal area of study, specifically of lipid pathways, including the qualitative and quantitative study of cellular lipid metabolism. Lipid dysregulation has been demonstrated in a variety of pathologies, such as cardiovascular disease and cancers including breast, prostate, colon, squamous cell cancer, melanoma, and more [6,7,8,9]. Disrupted lipid homeostasis has been discovered in glioblastoma, with over 500 lipids identified in several glioblastoma subtypes [10]. An understanding of serum lipidomics may offer ways to ascertain lipid signatures that help with glioblastoma diagnosis, prognosis, and treatment options. Here, we review lipid metabolism in both typical health and glioblastoma pathology, review the literature on lipidomics as potential biomarkers for diagnosis, prognosis, and therapy, and elaborate on future directions.

## 2. Lipid Metabolism in Health

Lipids provide a crucial energy source for the human body, serve as a structure to the cell membrane, and impact a variety of biologic pathways. They are classically insoluble in polar environments and soluble in organic solvents, and can be categorized as fatty acids, phospholipids, or neutral lipids, such as triglycerides and cholesterol esters. Fatty acids generally consist of a combination of hydrocarbons and a carboxyl group at the end. These fatty acids can be sub-categorized based on their length, the degree of double bond formation or saturation, wherein saturation signifies an absence of double bonds, and unsaturation consists of one or several double bonds, and the orientation of hydrocarbon chains around a double bond (i.e., *cis* or *trans*). Fatty acids play a number of roles in the body, including acting as structural components for cell membranes, influencers of tissue metabolism, energy storage, and fuel when alternative sources of energy, such as glucose, may not be readily available. The most fundamental structure of phospholipid includes the glycerol backbone, two hydrophobic fatty acid tails consisting of a hydrocarbon chain of varying lengths and degrees of saturation, and a hydrophilic phosphate group. Phospholipids are critical in cell membrane architecture, and allow for a dynamic amphipathic structure for cell protection and selective molecular transport. Triglycerides differ in that they consist of glycerol, bound to three fatty acid molecules. They predominantly serve as energy sources and support the absorption of fat-soluble vitamins. Cholesterol esters are the storage form of cholesterols, and differ from cholesterols in that there is esterification between the hydroxyl group of the steroid and fatty acid in a cholesterol. 

Anabolic pathways for fatty acid synthesis can begin in a number of ways, and here we highlight major mechanisms. The first is via glycolysis, whereby glucose is taken into the cell and ultimately synthesized into pyruvate, which can be converted to acetyl-coenzyme-A (acetyl-CoA) via pyruvate dehydrogenase and enter the tricarboxylic acid (TCA) cycle in the mitochondria. Other sources of acetyl-CoA include acetate, amino acids, and exogenous fatty acids which can all enter the TCA cycle. The TCA cycle, also known as the Krebs cycle, serves as an energy synthesizer by oxidizing a series of molecules to provide adenosine triphosphate (ATP), which is a major fuel source for the body.

One of the intermediates formed in the TCA cycle, citrate, can be shuttled out of the mitochondria, converted into acetyl-CoA, and synthesized into cholesterol and steroids, via a series of steps, or fatty acids, such as palmitate, which can further be metabolized into monounsaturated fatty acids (MUFAs), polyunsaturated fatty acids (PUFAs), phospholipids, and triglycerides. These can be exported throughout the body or stored in lipid droplets (LDs). LDs are composed of a hydrophobic core of lipids, an outer phospholipid layer which has various proteins interlaced throughout the phospholipid layer. These structures have long been considered storage organelle, but it is becoming more established that they are multi-functional and act as regulators of metabolism, cellular communication, and homeostasis [11,12,13]. 

Sterol regulatory element binding protein-1 (SREBP-1) is a master transcription factor involved in lipid homeostasis, including fatty acid and cholesterol synthesis. This transcription factor is composed of a helix–loop–helix leucine zipper and has a SREBP-1a, SREBP-1c, and SREBP-2 isoform. SREBP-1 is synthesized in the endoplasmic reticulum and combines with SREBP cleavage-activating protein (SCAP) to form a complex that allows for transportation to the Golgi apparatus. Upon arrival to the Golgi apparatus, it is cleaved sequentially by two proteases, site-1 protease and site-2 protease, to release the active SREBP, which can go to the nucleus for lipogenesis gene transcription [14]. The activated SREBP-1 exerts influence across de novo fatty acid synthesis enzymes, including, but not limited to, ATP citrate lyase, acetyl-CoA carboxylase, fatty acid synthase (FASN), and stearoyl-CoA desaturase 1 (SCD-1). SCD is critical for the synthesis of MUFAs [15,16]. SREBP-2 allows for the direct transcription of genes required for cholesterol biosynthesis and the uptake of cholesterol from exogenous sources [17]. 

Additionally, FASN is a large multi-functional enzyme that contains a monomer capable of synthesizing palmitate yet is unable to, since only its dimer form is functional [18]. It acts predominantly in the liver and adipose tissues, and is involved in many key processes, including embryonic and neural stem cell development. Pertinently, FASN is involved in propagating fatty acid synthesis reactions to produce palmitate through using malonyl-CoA as a substrate.

When energy is required, beta fatty acid oxidation can generate an immense amount of ATP when compared to similar catabolic processes from other energy storesm such as glycogen. Through a series of reactions whereby stored fatty acids are shortened by two carbons per cycle, fatty acid oxidation generates nicotinamide adenine dinucleotide (NADH) and flavin adenine dinucleotide (FADH2), which can proceed to the electron transport chain to generate ATP via oxidative phosphorylation. Acetyl-CoA is also a product of fatty acid oxidation, which can re-enter the TCA cycle to generate more NADH and FADH2 for ATP synthesis.

While this form of beta oxidation predominantly occurs in the mitochondria, peroxisomes can also process fatty acids, particularly very long chain fatty acids (VLCFAs) and branched chain fatty acids. In peroxisomal beta oxidation, fatty acids are not broken down completely, but reduced to a carbon chain length of 6-8 carbons that can then be shuttled to the mitochondria via carnitine to complete the oxidation process. Additionally, electrons that would otherwise go to the synthesis of FADH2 are instead donated to molecular oxygen in order to produce hydrogen peroxide.

## 3. Lipid Metabolism in Glioblastoma

Though the Warburg Effect highlights cancer cells’ ability to manipulate metabolic pathways, it focuses predominantly on the ability of cancer cells to preferentially increase lactate production, despite aerobic conditions [19,20]. 

It is now clear that cancer cells additionally transform other cell pathways, including the pentose phosphate pathway, to ensure cell survival [21], amino acid metabolism for neoplastic growth [22], increased growth factors for vascular growth [23], and, most relevantly, “lipid metabolic reprogramming”, wherein membrane synthesis, energy metabolites, and cell signaling are altered to support tumorigenesis and proliferation [24]. Ultimately, neoplasms, including glioblastoma, require an energy source, and lipids offer a high impact resource for survival. 

In glioblastoma, there is a great deal of heterogeneity in its cellular composition. Small RNA sequencing studies have indicated over 400 unique cells in glioblastomas that have varying degrees of transcriptional and, ultimately, phenotypic expression [25]. Eisenbarth et al. recently summarized the vast spectrum of sub-types amongst glioblastoma macrophages, T-cells, endothelial cells, glioma stem cells, myeloid-derived suppressor cells, fibroblasts, and tumor cells [26]. 

This heterogeneity is both linked to and independently seen in glioblastoma lipid composition and metabolism [27]. For instance, a mitochondrial subtype of glioblastoma was recently reported, whereby it showed more microglia-like cells and exclusively relied upon oxidative phosphorylation [28]. Additionally, enhanced lipogenesis and LD content have been demonstrated in several studies, with key findings detailed in Table 1 [29]. While the literature continues to evolve, a few key elements of lipid pathways have been identified as contributors to gliomagenesis and growth. Specifically, epidermal growth factor (EGFR) dysregulation has been associated with approximately 60% of glioblastoma cases [30]. EGFR is a tyrosine kinase receptor that is physiologically expressed in multiple organs, but commonly implicated in many pathologies. The most common EGFR mutation in glioblastomas is the EGFR variant III+ (EGFRvIII+), which contains deletions of exons two through seven, or a total of 275 amino acids, rendering it constitutively active [31]. EGFRvIII+ not only elevates the uptake of glucose, and offers protection from apoptosis, but it also allows for an increase in the glycosylation of SCAP, which, as aforementioned, chaperones SREBP from the endoplasmic reticulum to the Golgi apparatus, allowing SREBP to regulate lipid and cholesterol synthesis [32]. EGFRvIII+ and SREBP have been observed, and indicate that these early stages of fatty acid synthesis may be areas of future research [33,34].

ETS transcription factor 4 (ELF4) has been less studied than EGFR, but has been implicated in lipid homeostasis. ELF4 is elevated in high grade gliomas, particularly in glioma and neuronal stem cell markers, including CD44, CD36, CD15, CD70, S100A4, and ALDH1A3 [35]. The suppression of ELF4 interrupted lipid transportation by downregulating various lipid efflux genes and altering phospholipid levels, notably phosphocholine and phosphatidylethanolamine [35].

The lipid profiles of glioblastoma have demonstrated elevated FASN in glioma tissue and, specifically, in extracellular vesicles (EVs), including CD63- and CD8-positive glioblastoma EVs [36,37]. In glioblastoma cells, EVs have been found to promote angiogenesis, immune suppression, invasion, migration, and drug resistance [38]. The lipid composition of glioblastoma exosomes has also been characterized by increased sphingomyelin and saturated lipids, and elevated sphingosine-1-phosphate (S1P) has been found in glioblastoma tissues, suggesting that sphingolipids play a role in the spreading, invasion, and angiogenesis of tumors [39,40,41,42]. 

MUFAs, such as oleic acid, have also been associated with glioblastoma proliferation by increasing diacylglycerol-acyltransferase (DGAT) 1 expression, which is involved in triglyceride synthesis [43]. DGAT 1 is a transmembrane protein embedded in the endoplasmic reticulum. It esterifies acyl-CoA with diacylglycerol (DAG) to finalize triglyceride formation and ultimately to allow for the triglyceride to exist as a core component of LDs or lipoproteins [44]. DGAT 2 is another transmembrane protein in the endoplasmic reticulum with similar roles as DGAT 1, but others have reported that the two also action discrete functions. In mouse models, not only did DGAT 1 finalize triglyceride formation for storage, but it also protected the endoplasmic reticulum from lipotoxic impacts from exogenous sources, such as high-fat diets [45]. However, the same study also found that DGAT 2 had a greater role in triglyceride storage when compared to DGAT 1. In glioblastomas, DGAT 1 has dually demonstrated overexpression and was more efficacious when targeted, compared to DGAT 2 [43,46]. Future research into the structural and metabolic differences of DGAT 1 and 2 may elucidate tailored findings for its implications in glioblastoma cell lines.

Not only can neoplastic cells increase exogenous lipid uptake and lipogenesis, but they have also shown an increased ability to store lipid and cholesterol esters inside the specialized intracellular vacuoles of LDs [47]. Prior to the genesis of lipidomics, cholesterol esters had already been established as a specific element of glioblastoma [48]. This is due to the established understanding that brain cholesterol is largely synthesized de novo, as peripheral cholesterol cannot readily cross the blood–brain barrier [49]. As neurons and astrocytes generate oxysterols in the process of cholesterol metabolism, these oxysterols bind to liver X receptors (LXRs) to reduce extraneous cholesterol by reducing cholesterol uptake (i.e., LDL) and actioning the transportation of cholesterol out via various sterol transporters, such as ATP binding cassette A1 (ABCA1) [50]. The system not only works to reduce extraneous cholesterol, but also suppresses the rate-limiting enzyme of sterol genesis, HMG-CoA reductase. Emerging studies indicate that cholesterol is associated with glioblastoma survival, with LXR as a key receptor which regulates cholesterol levels [51]. The theory is linked to two types of study. The first is that LXR agonism has demonstrated CNS side effects in healthy individuals [52]. The second is the association of LXR as a promising target for other neoplasms, such as breast, colon, skin, or prostate, as well as squamous cell cancers [53,54,55]. In glioblastoma, LXR agonism has been linked to reduction in glioblastoma progression [51,55].

Additionally, sterol O-acyltransferase 1 (SOAT-1) impacts cholesterol ester formation in LDs. Normally, SOAT-1 is a transmembrane protein in the endoplasmic reticulum, which acts as a catalyst for the esterification of a long chain fatty acyl-CoA and intracellular cholesterol to form cholesterol esters that can reside in LDs for storage or in lipoproteins that can be transported elsewhere. By storing lipid and cholesterol esters inside LDs, glioblastoma cells can have ready access to energy stores for ATP production, and are able to prevent lipotoxicity and blunt cell death, including ferroptosis [56,57]. The amplification of SOAT-1 has similar effects to DGAT-1/2, as they are both involved in lipid storage and propagate glioblastoma aggressiveness [43,46,58]. 

LDs have been found in glioblastoma organoid cores and nutrient deficient pseudopalisading and perinecrotic areas of glioblastoma cells, suggesting that LDs may serve as an energy supply for glioblastoma cells in these areas where they might not otherwise have a rich environment for proliferation [59]. These findings are consistent with studies that show how LDs critically contribute to cell survival during starvation [13]. Interestingly, exogenous fat from high-fat diets were associated with fatty acid accumulation in tumors that, in turn, promoted glioma growth and protection from necrotic cell death by inhibiting hydrogen sulfide [60].

Lipid catabolism by way of fatty acid oxidation in glioblastoma differs compared to healthy cell lines. Acyl-CoA-binding protein (ACBP), otherwise known as a diazepam binding inhibitor (DBI), normally regulates neural stem cell proliferation by interacting with the GABA-A receptor. ACBP is elevated in glioblastoma, effectively allowing glioblastoma cells to proliferate [61,62]. In its multifunctional role, ACBP also binds to medium- and long-chain acyl-CoA esters intracellularly, indicating a role in lipid metabolism. Particularly, increased ACBP expression allows for glioblastoma cells to proliferate with a supplementary source of energy.

Another set of key lipid catabolism enzymes implicated in glioblastoma include carnitine palmitoyltransferase (CPT-1) isoforms a and c, CPT-2, and acyl-CoA dehydrogenase 9 (ACAD9) [43]. CPT-1 is located in the outer mitochondrial membrane and catalyzes the transport of long-chain fatty acids into the mitochondria for beta oxidation. CPT-2 performs similar functions, but resides inside the mitochondria. ACAD9 assists in the assembly of complex I of the electron transport chain and the catalysis of acyl-CoA’s to 2,3 enoyl-CoAs in fatty acid oxidation. These aforementioned fatty acid oxidation enzymes have been mutually enhanced with CD47, an immune checkpoint receptor that protects cycles from phagocytosis [63]. The contribution of the enzymes enumerated may serve as targets of future research for therapy. 

**Table 1 cancers-16-01089-t001:** Molecules implicated in glioblastoma lipid dysregulation.

Title 1	Expression	Implication	References
EGFRvIII+	↑	Increased fatty acid synthesis	[32,33,34]
ELF4	↑	Increased lipogenesis and cholesterol synthesis	[35]
Extracellular vesicles	↑	Increased sphingomyelin, saturated lipids, and sphingosine-1-phosphate	[39,40,41,42]
DGAT 1/2	↑	Increased MUFA/triglyceride synthesis	[43]
SOAT-1	↑	Increased lipid storage and cholesterol ester formation in lipid droplets	[43,46,64]
ACBP	↑	Increased fatty acid oxidation	[61,62]

ELF4 = ETS transcription factor 4; DGAT = diacylglycerol-acyltransferase; MUFA = monounsaturated fatty acid; SOAT-1 = sterol O-acyltransferase-1; ACBP = acyl-CoA-binding protein.

## 4. Serum Lipidomics in Glioblastoma Diagnosis and Prognosis

Studies indicate that the lipid profile may be used to support the diagnosis of glioblastoma. Universal lipid profiling in a small sample size study showed that lipid classes including fatty acids, glycerophospholipids, and glycerolipids had significantly different representations when compared to non-glioblastoma controls [65]. LDs and SOAT-1 have also been proposed as potential biomarkers, given their overexpression in glioblastoma [58]. A 2021 rapid evaporative ionization mass spectrometry study showed higher intensities in 12 lipid metabolite biomarkers in glioblastoma, relative to healthy tissues. These biomarkers included various fatty acids and phospholipids, many of which were consistent with other studies that had similar findings [66].

The utility of lipid biomarkers may help in the early detection and monitoring of glioblastoma. In other cancers, FASN overexpression begins in the precancerous stage and persists through the course of the disease [67]. Glioblastoma presents a challenge as it can have a poor prognosis from the outset and involve sampling strategies that involve more risks with repeated occurrence, such as brain biopsy and CSF extraction. In order to determine lipid signatures in glioblastoma on a wider and less invasive scale, a recent study used extracted lipids from plasma samples and support vector machine-based learning, liquid chromatography, and mass spectrometry to find 11 potential biomarkers, including lysophosphatidylcholines, phosphtidylcholines, and triglycerides [68]. 

In addition to early detection, it may be possible for lipids to distinguish markers between glioma subtypes. For instance, in isocitrate dehydrogenase-1 (IDH1) mutant gliomas, decreased triglycerides and sphingolipids have been demonstrated to be fewer in quantity relatively to IDH1 wild type gliomas, which may have been attributed to lower levels of long-chain acyl-CoA synthetase (ACS) 1, ACS4, and very long-chain ACS3 [68]. This level of difference may help in diagnosis and tailored therapy. 

However, it is important to emphasize that these are the early stages of research and warrant vigilant testing. Though early literature first proposed that 24S-hydroxycholesterol could be a lipid that served as a serum biomarker for brain tumors, this proved to be more apparent in trauma rather than central nervous system neoplasms [69,70]. The area of potential biomarkers remains understudied, and future research using methods, including computational approaches, may be productive [66,71]. 

In addition to diagnosis, serum lipidomics may be fruitful for prognosis. The overexpression of polymerase 1 and transcript release factor (PTRF), also known as Cavin1, has been linked to the malignancy grade and worse prognosis in glioma patients [72]. While PTRF contributes to the formation of caveola in the plasma membrane, a follow-up study showed that PTRF also activates the cytoplasmic phospholipase A2 (cPLA2) pathway, which ultimately remodels phospholipids in vivo and reduces CD8+ tumor infiltrating lymphocyte production in response to glioblastoma [73]. These findings indicate that PTRF may not only be helpful in glioma diagnosis, but also in gauging prognosis.

## 5. Serum Lipidomics as a Therapeutic Target

Glioblastoma has proved a dilemma for treatment due to its heterogeneity and plasticity that confer therapeutic resistance [74]. Temozolomide (TMZ) is the standard chemotherapeutic agent in glioblastoma treatment, but its therapeutic capabilities are often short-lived. Herein, a precise understanding of the lipid profile of glioblastoma can offer an avenue for targeted therapy, with the key therapeutic targets summarized in Table 2. Choo et al. propose that TMZ resistance may be linked to lipid composition [75]. TMZ-resistant glioblastoma cells were found to have higher cholesterol and fatty acid synthesis via the overexpression of SRBEP and lower lipid unsaturation with increased membrane rigidity. Lipid anabolism inhibitors such as fatostatin, a SREBP inhibitor, demonstrated the inhibition of TMZ-resistant glioblastoma cells compared to TMZ-sensitive cells. In this way, it is possible for TMZ treatment to be more efficacious when combined with therapies that target the key players involved in lipid biochemical pathways [75].

Not only is this true for fatostatin, but several studies have also demonstrated that SCD and fatty acid desaturase (FAD) inhibition activates lipotoxicity. This occurs by preventing the formation of MUFAs and, when combined with TMZ, additionally impacts DNA damage repair [27,76].

A growing number of studies are evaluating target lipid metabolites to stunt glioblastoma progression. The inhibition of Acetyl-CoA carboxylase (ACC) has been found to inhibit the EGFRvIII+ capacity to proliferate and alter lipid content in glioblastoma [34]. Indraccolo et al. recently showed that regorafenib can phosphorylate ACC and can thereby inhibit ACC activity, subsequently affecting EGFRvIII+ glioblastoma cell growth [77].

Fatty acid synthesis has also received interest. Cerulenin, a fatty acid synthase inhibitor, was found to decrease fatty acid synthesis by 50%, and the RNAi-mediated knockout of FASN demonstrated apoptosis induction and the accumulation of cells in the S phase of the cell cycle [36]. SOAT-1 inhibitors, such as avasimibe, have also blocked LD formation and SREBP-1-Mediated FASN [64]. 

With the knowledge of glioblastoma’s reliance on cholesterol as a nutrient source, the mevalonate pathway has been investigated as a potential target for therapy. Lovastatin and phenylacetic acid have been found to synergistically suppress glioblastoma growth, by inhibiting HMG CoA reductase and MVA-pyrophosphate decarboxylase, respectively, and the increased expression of peroxisome proliferator-activated receptors, which are integral for processes of lipid metabolism, cell growth, and differentiation [78].

LXR agonism has also demonstrated promise as a mechanism to destroy glioblastoma cells [55]. In particular LXR-623 has been shown not only to successfully penetrate the blood–brain barrier, but also to induce glioblastoma regression and prolonged survival [51]. 

Another therapeutic target for glioblastomas is focusing energy on elements that offer glioblastomas protection from lipotoxity. DGAT 1 is a prime example that has continued to show that, in addition to increasing energy storage, it also protects glioblastomas from oxidative damage [46]. Focusing on DGAT 1 inhibitors in concert with lipogenesis inhibitors may be a fruitful approach to reduce glioblastoma activity and advancement.

In animal models, etomoxir, a fatty acid oxidation inhibitor, has been found to alter glioblastoma cells’ ability to survive in nutrient-poor environments [79]. Not only does this reiterate lipids as a robust energy source for glioblastoma, but it also highlights potential therapeutic targets.

While these studies have exhibited a degree of experimental success, it remains unclear how targeting enzymes that are involved in normal human physiology may impact patient treatment. Additionally, the blood–brain barrier poses an interesting dilemma as it acts as a tight gatekeeper for molecular transport. Lipids may be a way by which therapeutic agents can be delivered as they can cross the blood–brain barrier via diffusion. A murine study found combinations of epirubicin, and resveratrol integrated into liposomes to have better transport across the blood–brain barrier and higher survival rates in mice with gliomas [80]. Similar success has yet to be reached in humans. A study in 2004 found a modest 6-month progression free survival (PFS) to be 32% [81]. A phase II trial of TMZ and pegylated liposomal doxorubin for glioblastoma conducted in 2011 found improved transport across the blood–brain barrier, but no significant benefits on 6-month progression free survival or survival overall [82].

## 6. Future Directions

As the field of lipidomics is still in its early stages, additional research across the spectrum of glioblastoma, from diagnosis, prognosis, and targeted therapies, would be beneficial. There is particular promise in the discovery of lipid biomarkers that are reliable as early biomarkers, or perhaps that can denote risk factors of disease. Though certain metabolites, such as fatty acid synthase and SREBP, currently hold a lot of attention, tens of thousands of lipid species exist. These lipid molecules require elucidation in terms of structure and function. There may be benefits in combining clinical and scientific expertise with artificial intelligence (AI). As AI is a rapidly becoming available to the public and experiencing growth as a tool, utilizing this technology may be a way to expedite our understanding of lipids and their connection to glioblastoma. 

## 7. Conclusions

Recent research in glioblastoma lipidomics has resulted in a number of fascinating discoveries that highlight the unique composition of glioblastoma cells. It is clear that lipids play an integral role in glioblastoma growth and proliferation, and there is opportunity for these metabolites and pathways to be understood with the collaboration of advancing technology. Our knowledge of the lipidome is ever increasing, and it is critical that this field of research illuminates markers for early diagnosis, the accurate discernment of prognosis, and targets for glioblastoma therapy.

## Figures and Tables

**Table 2 cancers-16-01089-t002:** Therapeutic trials targeting lipids in glioblastomas.

Target	Implication	References
SREBP	Inhibition; decrease in TMZ resistance, decreased MUFA formation	[27,75,76]
Acetyl-CoA carboxylase	Inhibition; decreased EGFRvIII and glioblastoma cell growth	[34,77]
Fatty acid synthase	Inhibition; apoptosis	[36]
SOAT-1	Inhibition; decreased lipid droplet formation and fatty acid synthase	[64]
HMG CoA reductase	Inhibition; decreased glioblastoma growth	[78]
MVA pyrophosphate decarboxylase	Inhibition; decreased glioblastoma growth	[78]
CPT-1	Inhibition; decreased fatty acid oxidation	[79]
Liposome	Increased transport across the blood brain barrier	[80,81,82]

SREBP = Sterol regulatory element binding protein-1; SOAT = sterol O-acyltransferase-1; TMZ = temozolamide; CPT-1 = carnitine palmitoyltransferase.

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
