# Peer review of "The Lipidomic Signature of Glioblastoma: A Promising Frontier in Cancer Research"

_cancers, 2024, doi:10.3390/cancers16061089_

Round 1

Reviewer 1 Report

Comments and Suggestions for Authors

This review is interesting and well organized on the complex field of Glioblastoma research.

Some lines of English language could be improved (e.g. “lens by which to view glioblastoma” (P1,L37); “immune amount” (P3,L108); “while literature continues to grow” (P3,L132-133); “include tumorigenesis” (P3,L137).  "AI" on P7-8, lines 337-340; I do not think that AI would expedite our understanding of lipids.

Comments on the Quality of English Language

Please see  "Comments and Suggestions for Authors
" box.

Reviewer 2 Report

Comments and Suggestions for Authors

Glioblastoma (GBM), categorized as a WHO grade 4 astrocytoma, represents approximately 14.3% of all primary brain and central nervous system neoplasms, making up a substantial 49.1% of malignant brain tumors. It is associated with a bleak prognosis, characterized by a median survival of just 14.6 months following diagnosis. This review provides a comprehensive exploration of potential lipid pathways and proposes future avenues for lipidomic research in glioblastoma.

Additionally, it would be beneficial for the authors to briefly outline targeted drugs associated with each lipid pathway, offering insights into current or potential therapeutic options for glioblastoma patients.

To support their assertions, the authors could include the following papers: PMID: 31468706 at line 184 and PMID: 31468706 at line 302.

Reviewer 3 Report

Comments and Suggestions for Authors

The manuscript "The Lipidomic Signature of Glioblastoma: A Promising Frontier in Cancer Research" by Nina Yu and Orwa Aboud is a well written review of the state of the art of lipidomics in glioblastoma that highlight almost all the important concepts in the field. One exception is the lack of discussion of the importance of heterogeneity in cellular composition of GBM in vivo and the possible effects that has on the lipid asset of the tumor. Another consideration, with ample room for improvement, is that sometimes the authors do not indicate the original paper reporting the seminal finding but they provide more recent references or indicate a review. An example is the indication that the mevalonate pathway may be targeted for therapy where the one of the original references is missing1.

Minoir Points

pg 6 rows 288-9 Please provide the appropriate references to the sentence "Lipid anabolism inhibitors such as fatostatin, a SREBP inhibitor, demonstrated inhibition of  TMZ-resistant glioblastoma cells compared to TMZ sensitive cells.

REFERENCE

1) Soma MR et al. In vivo enhanced antitumor activity of carmustine [N,N'-bis(2-chloroethyl)-N-nitrosourea] by simvastatin. Cancer Res. 1995 Feb 1;55(3):597-602.

Reviewer 4 Report

Comments and Suggestions for Authors

The review by Dr Nina Yu and Orwa Aboud, entitled “The Lipidomic Signature of Glioblastoma: A Promising Frontier in Cancer Research”, addresses a very interesting novelty in glioblastoma research consisting in using lipidomics to study the altered lipid metabolism occurring in glioblastoma cells in order to perform diagnosis and plan new therapies to fight back against this devastating brain tumor.

The review examines with property and in depth the issue of the dyslipidemia that occurs in glioblastoma, pointing out the future perspective of using this approach to improve diagnosis, prognosis, and develop targeted therapies.

The review is written very well and addresses the topic with competency. I have only a minor points, which are the following:

-Page 3, line 124: Apart from the Warburg Effect mentioned here, explain in a more extensive way that cancer cells are able to re-program many other cell pathways, for example stimulating the production of growth factors that promote the formation of intratumoral vessels.

-A schematic figure (i.e., an illustration) showing the main lipid pathways that are altered in glioblastoma would be useful to better comprehend the metabolic dysregulation described in the subsection 3.
